# From Urban-Scape to Human-Scape: COVID-19 Trends That will Shape Future City Centres

**Elizelle Juanee Cilliers** [1,2,*] **, Shankar Sankaran** [1] **, Gillian Armstrong** [1] **, Sandeep Mathur** [3] **and Mano Nugapitiya** [4]

1 Faculty of Design, Architecture and Building, University of Technology Sydney, Ultimo, Sydney 2007, Australia; Shankar.Sankaran@uts.edu.au (S.S.); Gillian.Armstrong@uts.edu.au (G.A.)
2 Unit for Environmental Sciences, North-West University, Potchefstroom 2531, South Africa
3 Transport for NSW, Sydney 2000, Australia; Sandeep.Mathur@transport.nsw.gov.au
4 Ontoit Global Pty Ltd., Sydney 2000, Australia; Mano.Nugapitiya@ontoit.com
* Correspondence: jua.cilliers@uts.edu.au

**Abstract:** The COVID-19 pandemic did not only impact all spheres of life but came abruptly to redefine our understanding of the urban-scape. With changing user-values and user-needs, there is a renewed realisation of the importance of the human-scape and how human capital, social issues, and liveability considerations will progressively lead urban development discussions. The urban-scape risk is far more complex and fragile than previously anticipated, with the future of the city centre dependent on our ability to successfully manage the transition from an urban-scape to a human-scape. This research employed a narrative review methodology to reflect on COVID-19 trends that will shape future city centres, based on expert contributions pertaining to (1) the community sector, (2) the public sector, and (3) the private sector within the Sydney Metropolitan area of Australia. The research highlighted the changing human-scape needs and associated impacts of (1) changing movement patterns, (2) changing social infrastructure, and (3) increasing multifunctionality, which will be crucial factors in shaping attractive (future) city centres. The research contributes to the notion that future city centres will embrace and prioritise the human-scape in a response to 'build back better', and accordingly, identified how the human-scape can be articulated in broader spatial planning approaches to create attractive future city centres.

**Keywords:** urban-scape; COVID-19; cities; changing landscape; social capital; public transport; university campus; private business; communities; Sydney

## 1. Introduction: COVID-19 Changing Business-as-Usual for Cities and Sectors

The fragility of the city centre has recently been highlighted by the COVID-scape and the broad range of challenges and changeable unknowns it abruptly introduced. Changing societal needs and associated work-life patterns have emerged on a global scale as a result of the worldwide COVID-19 pandemic response and lockdown restrictions put in place. The impacts of the COVID-19 pandemic have introduced various challenges, but also opportunities for communities and future development efforts [1]. It has emphasised that social issues will progressively lead urban development discussions and echoed the findings of previous research that stated that the balance between the natural and built environments will continuously be negotiated in terms of liveability considerations [2]. During the past 18 months, cities across the globe have witnessed how the human-scape has drastically changed to conform to new ways of working and moving to such an extent that some scholars are now questioning the future of the city centre [3]. The contemporary urban-scape is being challenged by the current COVID-19 trends, and while the risks of the feasibility of the contemporary city centre are being highlighted, the current trends also provide a unique opportunity to reflect on the role of the urban-scape, the changing human-scape, and how we can transition to create attractive future city centres.

This research reflected on the trends and impacts of COVID-19 responses based on three different perspectives, informed by expert contributions with lived experiences in the Australian metropolitan city of Sydney from respective stakeholder groups inclusive of (1) the community, (2) the public, and (3) private sectors, as elaborated accordingly.

### 1.1. The Community Sector, a University Campus

This perspective includes the views of an expert with lived experiences, working in the community sector in Sydney and reflecting on the impacts of the COVID-19 pandemic on the higher education environment as applied to the University of Technology Sydney (UTS), located in the city centre of Sydney. The city campus is located near Sydney Central Station in Sydney and spread over five precincts, with the main campus located in Broadway, Haymarket, and Blackfriars, a sports hub in Moore Park, and an industrial centre in Botany close to the international airport. The location of UTS enhances its accessibility and reach and is a reason why UTS enrols close to 46,000 students each year from across the globe. This community sector saw an abrupt change in March 2020, when the COVID-19 pandemic started, and the university initiated a campus-wide lockdown. Face-to-face teaching transitioned to online and virtual learning, as business-as-usual was not possible under the restrictions on movement and social interaction The location coefficient that benefitted UTS was no longer part of the equation for attracting students, and industry collaboration and student experiences had to be redefined to fit the virtual environment. Changing work-life patterns and new ways of working (and learning) highlighted the role that the city centre previous played to sustain this community sector, but also raised questions relating to the future role of the city centre and whether it would be valued for its agglomeration economies alone.

### 1.2. The Public Sector, a Public Transport Network

This perspective includes the views of an expert with lived experiences, working in the public sector in Sydney and reflecting on the impacts of the COVID-19 pandemic on the Transport for NSW (New South Wales, Australia) network. Transport for NSW leads the development of a safe, efficient, integrated transport system that keeps people and goods moving, connects communities, and shapes the future of our cities and regional centres. It is responsible for strategy, planning, policy, regulation, funding allocation, and other non-service-delivery functions for all modes of transport in NSW including road, rail, ferry, light rail, point to point, regional air, cycling, and walking. With over 25,000 people working across NSW, it focuses on improving the customer experience and contracting public and private operators to deliver transport services on its behalf. This public sector has experienced the impacts that the COVID-19 pandemic brought along, specifically in terms of the vastly adjusted movement patterns. The city centre, which used to be the central connection hub with linking transport corridors and supporting infrastructure and services, was forced to a near standstill as part of lockdown regulations. This public sector's focus shifted from moving large groups of people as efficiently as possible to moving large groups of people as safely as possible, providing flexibility and choice through public infrastructure.

### 1.3. The Private Sector, an Advisory Consultancy for Built Environment Infrastructure

This perspective includes the views of an expert with lived experiences, working in the private sector in Sydney and reflecting on the impacts of the COVID-19 pandemic on the Ontoit Global Pty Ltd. Group (Sydney, Australia). Ontoit is a medium-sized independent infrastructure advisory and project management consultancy services business with over 70 staff members. It delivers services across nine different service sectors and has both public and private sector clients. Their consultancy work supports the major infrastructure market along the eastern seaboard of Australia [4]. Ontoit has offices in the major cities along the eastern seaboard of Australia, which are all strategically located close to transport and entertainment precincts. The COVID-19 pandemic shifted the Ontoit approach from

face-to-face interaction (which was always considered a strong contributor to business culture, particularly in multidisciplinary organisations) to online and hybrid interaction. During the COVID-19 pandemic, this private sector had to adapt to virtual interaction and new ways of working, which in some instances were considered to be more beneficial in terms of time and resource management, thus questioning the future of the city centre as economic hub, which to some extent could be mimicked in the virtual environment.

## 2. Literature Review: The Notion of Urban-Scape and Human-Scape

Urban-scape dynamics have always been shaped by complex interactions across social, economic, and political factors, including population distribution, flows of wealth, and infrastructure requirements. As a result, city centres embodied a sense of unique human entrepreneurship, economic dynamism, and evolving multiculturalism [5]. The COVID-19 pandemic disrupted the urban-scape and the very core of urban living, from social distancing measures to lockdown responses, to destabilized local economies [6]. City centres are now challenged to respond in an attempt to reclaim the attractive urban-scape, along with its vibrant, humanized communities. It is no easy task, as the contemporary role and function of city centres have been challenged by technological advances [5] and communication innovations which now offer tantalising possibilities for transcending traditional social and geographical barriers [7]. These new advances in communication technologies and supporting metropolitan transport systems allows citizens to now stay selectively in touch, while disconnecting from the city at large [8]. This brings along a totally new understanding of efficiency (costs) and justice (equity), which underpins the performance of the contemporary city. It reopens the discussion of the five basic dimensions for the performance of a city, drafted by Lynch [9] namely: (1) how form affects vitality, (2) how form affects human sense, (3) the degree to which the form fits the requirements of people, (4) how people are to access activities and services, and (5) how much control people have over services, activities, and spaces. The COVID-19 pandemic reiterated the importance of the human-scape that was acknowledged by Lynch (1981) [9] but became overshadowed by agglomeration economies and related technology-driven production processes of the contemporary city centre [5]. The COVID-19 pandemic also brought forth a new understanding of changing societal needs and highlighted the fragility of cities when the human-scape is disrupted. Research furthermore showed that urban fragility is a crucial consideration for modern cities that are committed to coping with growing external pressures (from the environment) and internal tensions (within the social system) [10]. Urban fragility would continue to play an even bigger part in managing post-COVID city centres with the growing awareness of the importance of the human-scape as an integral part of the urban-scape. The importance of the human-scape is no new phenomenon, drawing back to the work of Jacobs, who identified more than half a century ago what the potential and real failures will be when the human dimensions of the city were not considered, accounted for, and valorised [11]. Human needs are thus the core basis and foundation on which urban futures should be usefully envisaged, and the recognition of how the human-scape intertwines with the urban-scape underpins an appreciation of the growing complexity of city centres [10] and the different layers and interactions that need to be considered [12]. While human-scape refers to the social and cultural system, and how humans socially interact with their physical environment, the urban-scape relates to the physical environment or 'landscape' of the built environment. The relationship between the urban-scape and the human-scape manifests through the complementary and conflicting impacts that they have upon each other, through public health improvements, social cohesion, social equality, and economic systems measuring urban capital [10].

The challenge, however, is that the urban-scape is traditionally known to be a slow-changing environment, while its hosting societies, the human-scape, are increasingly becoming more dynamic [2]. A slow-changing urban-scape must respond to fast-changing needs of the human-scape, brought along by the increasingly dynamic (and changing) needs and preferences with regard to social, sustainability, and economic issues, and further

inflated challenges brought along by the COVID-19 pandemic. Recent developments emphasised the urgency of previous calls to foreground human needs at the heart of urban societal futures [11] and rethink the current policy and governance approaches that primarily focus on economic, technical, and environmental imperatives at national scales.

What is evident is the significance of urban identity, distinctiveness, and meaning that is increasingly being highlighted [13], especially when considering the transition from urban-scape to human-scape. Urban identities are informed by the "perceived uniqueness of a place" [14] and are formed by a range of "multiple and mutual relations in between context and content" [13]. The need for urban identity draws on a deep human need for associations with significant places [15]. Persistent identity has three distinctive components, namely the physical setting, the activities, and the experiences linked to that place [16]. The future cities of the post-COVID reality would need to revisit the notion of urban identity and uniqueness, and facilitate the transition from the urban-scape to the human-scape in a quest to reclaim the attractiveness city centres.

### 3. Contextualising Urban Identity: The Australian Context

From an Australian perspective, urban planning has progressively established a modern geospatial capability, which is known for its economic, social, and environmental well-being [17]. The human-scape has become increasingly recognised as part of the urban identity, and as a result, Australia is home to some of the most liveable cities in the world, according to the Global Liveability Index of 2021 [18], which evaluated 140 global cities based on a liveability score, inclusive of 30 qualitative and quantitative factors across five categories including stability, healthcare, culture and environment, education, and infrastructure.

The PwC Australia's Future of Work Report [19] showed that Australian city centres attract over 39% of staff employed by the professional and financial services industries who have a 'high' to 'very high' capability of being able to work remotely, as illustrated in Table 1.

With the introduction of COVID-19 and city-wide lockdown responses put in place in early 2020, the geospatial economic model also saw a fast change in terms of composition, since the traditional urban-scape could no longer provide the qualities that were needed to host and sustain the human-scape or business-as-usual [12]. The attractiveness of the city became much less important than attracting and retaining staff. Apart from the majority of professional and financial services industries retracting from the city centres, leaving the urban-scape without function or identity, other urban land-use changes were also evident during the time of the pandemic. Open space and quality outdoor environments within the urban-scape became crucial for well-being, and the benefits and attractiveness of these spaces were widely recognised in Australia, as in other global cities [20–22].

There was a renewed recognition of the importance of the urban-scape to provide social experiences [23], infrastructure, and spaces to support physical and mental health [24,25], flexible and safe transport modes [26], and access to digital infrastructure [27,28] as part of basic human rights for urban living and enhanced liveability. A new definition of essential tasks [29], the importance of transportation for global and geographically remote cities such as Sydney [30], changing travel patterns [30], and the decentralisation of the urban centres driven by the drastic change in user-values and user-needs [31,32] further challenged the contemporary city centres. Limited mobility, remote working, and prioritising the health agenda became the new normal within the urban-scape, turning planning theories and our understanding of the morphology of cities upside down [20]. Even urban density, which was previously not considered a risk factor [33], was now reconsidered as a driver of infection rates within the urban-scape [34], adding to the list of new vulnerabilities that were introduced as part of the 'new normal' in cities [35].

**Table 1.** Capacity of city centre workers with capability to work from home.

| Industry | % of CBD Employment | WFH Capability |
|---|---|---|
| Professional, Scientific, and Technical Services | 21.14% | High |
| Financial and Insurance Services | 17.63% | Very High |
| Public Administration and Safety | 13.90% | Low |
| Accommodation and Food Services | 7.33% | Very Low |
| Education and Training | 4.97% | Very Low |
| Retail Trade | 4.77% | Medium |
| Administrative and Support Services | 4.72% | Low |
| Information Media and Telecommunications | 4.17% | Medium |
| Health Care and Social Assistance | 4.01% | Very Low |
| Construction | 3.31% | Low |
| Rental, Hiring, and Real Estate Services | 2.46% | Medium |
| Mining | 2.45% | Low |
| Transport, Postal, and Warehousing | 2.26% | Low |
| Other Services | 1.81% | Very Low |
| Electricity, Gas, Water, and Waste Services | 1.67% | Low |
| Arts and Recreation Services | 1.52% | Very Low |
| Manufacturing | 1.04% | Low |
| Wholesale Trade | 0.86% | Medium |
| Agriculture, Forestry, and Fishing | 0.00% | Very Low |
| Total | 100% | |

Source: PwC's Geospatial Economic Model [19] .

Various attempts have been launched to limit the risks now posed to the human- and urban-scape, encapsulated as part of the City of Sydney's Recovery Plan [36] which was informed by inputs collected from a survey opened to public consultation. In an attempt to manage the complex interactions across social, economic, and political factors, and to understand the risk of the future urban-scape, more emphasis should be placed on understanding the human-scape and associated changes of user-values and user-needs, which will ultimately shape attractive future city centres [20]. Therefore, this research further investigated the human-scape, based on the perspectives of three different sectors, included as part of a narrative review methodology.

## 4. A Narrative Review: Reflections of Sydney's Urban Identity

This research employed a qualitative methodology, in the form of a narrative review, to understand the changing needs and values connected to the human-scape and associated urban-scape amid the COVID-19 global pandemic. The narrative review followed a phenomenological approach in an attempt to understand the social reality, grounded in people's experiences pertaining to the social reality [37]. The narrative review aimed to contextualise lived experiences and practices, capturing the human-scape through narratives, and drawing on the events and impacts brought forward by the COVID-scape.

To understand the changing human-scape needs and associated risks posed to the urban-scape, this research reflected on the impact of COVID-19 from three different perspectives: (1) the community sector, (2) the public sector, and (3) the private sector. The perspectives were captured by means of expert contributions of individual stakeholders sharing their reflections from each of the respective sectors. Expert contributions were selected as part of the methodology and the exploratory phase since it is proven to be a more efficient and concentrated method of gathering data than, for example, participatory

observation or systematic quantitative surveys. Expert contributions as part of empirical research are underpinned by the theory of society and a democratic theory perspective, as well as from the sociology of knowledge, scientific, or technical research standpoints [38,39]. Expert knowledge also pertains to modernization theory, where changes in the modern world form the point of view of knowledge dynamics, and is therefore part of "institutional reflexivity" [40]. This research relied on expert contributions to provide insider knowledge [38] and access to a particular social field related to change management during the COVID-19 pandemic, from the perspective of the community, public sector, and private sector. Expert contributions included perspectives of (1) the Faculty of Design, Architecture, and Building, UTS; (2) Transport for NSW; and (3) Ontoit Global Pty Ltd., Sydney, NSW, Australia. The qualitative data include reflections since the start of the COVID-19 pandemic and first impacts, which were observed in Australia in March 2020, with the analysis including the period until mid-2021, when the pandemic was still ongoing and events unfolding. The perspectives informed a thematic analysis and identified challenges and opportunities for shaping future city centres (amid the trends brought about by the pandemic).

## 5. Analysis: Contextualising the Challenges and Opportunities of the City Centre

The shift to online spaces and removal of physical boundaries had differing impacts upon the competitive positioning of sectors which have previously engaged with central locations. An overview of the analysis of the reflections drawn from the empirical investigation are provided in Table 2 below, followed by an exploration of the challenges and opportunities by: (1) the community sector, (2) the public sector, and (3) the private sector.

**Table 2.** Challenges and opportunities identified from the narrative review.

| Considering the Urban-Scape and Changing Human-Scape | |
| --- | --- |
| **Challenges Identified** | **Opportunities Identified** |
| Location no longer competitive advantage | Advanced communication technologies |
| Vibrant city identity no longer competitive | Greater flexibility of space |
| Inequality became more evident | High-value contact time |
| Increasing economic and social constraints | Meaningful engagements |
| Traditional workplace no longer deemed fit | Activity-based working possibilities |
| Traditional work ways no longer deemed fit | Active transportation modes |
| Health agenda dictated a new normal | Repurposed buildings |
| Limited agglomeration economics | Multipurpose spaces |
| Lack of attractiveness and sense of place | Creation of working hubs |
| Fast-changing societal needs | Evolving multiculturism |

### 5.1. Challenges Observed in the Human- and Urban-Scape

The location of the urban-scape was no longer considered a competitive advantage as digital shifts depicted a reduction in competitiveness across multiple physical locations.

(1) "Being located in the city centre was UTS's competitive advantage with convenient transport modes and connections, as well as availability of adequate student accommodation. COVID-19 impacted the locational advantage and replaced the central teaching location with a virtual learning environment that had no geographical boundaries. The urban-scape, as experienced by this sector, was transformed by the digital economy."

(2) "The traditional workplace and ways of working were no longer deemed fit when COVID-19 response measures were put in place. It became evident that hybrid

working approaches would be the way forward and that more focus would need to be placed on enhanced connectivity and workplace experience."

(3)　"Ontoit has progressively embraced cloud-based technologies to enable effective collaboration between staff located in its interstate offices and its remote clients. This enabled a rapid transition, enabling staff from remote locations, moving away from the economies of scale which the urban-scape previously offered."

The digital shift was accelerated and applied on a much larger scale than previously. This shift has had complex impacts on the competitiveness of cities, with expectations to pivot amid a changing landscape of public health information. Inequality also became more evident across boundaries.

(1)　"The vast inequalities that previously existed between different countries, regions, and areas became more evident when international students now had to attend classes from their home locations, in some cases not being able to access the technological advances and infrastructure which was usually at their disposal through the UTS campus. These challenges, which were previously not considered in the higher education environment in Sydney, such as efficient and accessible internet connections, data availability, and digital communication, were now part of the reality."

(2)　"The long-term planning and visions of Transport for NSW had to instantly adapt to cater for changing societal needs and an unknown future about the needs of the resident population, how movement and travel patterns would change over time and the role of public infrastructure in creating healthy, sustainable city centres."

(3)　"Traditional workplace and working ways were no longer deemed fit as the COVID-19 pandemic changed the face-to-face culture of Ontoit, abruptly reverting to remote working arrangements, with access to the offices limited by its COVID-19 safe working policy informed by federal and respective state government rules. This meant a complete shutdown of offices or limited attendance capacity to meet safe physical distancing regulations. With large businesses reverting to a similar arrangement, there was a complete desertion of city centres by office workers, which had a further abrupt and cascading impact on service-related businesses in the urban-scape."

All three sectors revealed that while digital technologies can enable a more streamlined and consistent experience for communities across Sydney, the vibrancy of the physical city was very much valued as a competitive and social equity advantage for all communities. The city-wide shift to a digital space has brought a complex range of impacts for all three communities.

(1)　"The vibrancy of the city centre, and the social and cultural benefits it provided as part of the community sector's support system, was now more evident than ever. The sector was challenged to respond to vast changing social needs underpinned by a health agenda."

(3)　"A large part of Ontoit's business-to-business engagement relied on its city centre location, nearby transport hubs, dining, and entertainment precincts in close proximity, which was no longer a competitive advantage, as the resident population migrated out of offices in the city centre to work remotely. Limited agglomeration economics, along with a lack of attractiveness and sense of place was characterising the urban-scape."

*5.2. Opportunities Observed in the Human- and Urban-Scape*

Advanced communication technologies were rapidly introduced which offered greater flexibility of space within the urban-scape.

(1)　"As UTS moved towards a blended learning mode to accommodate both online and on-campus students, greater flexibility was introduced, drawing on keeping the best of both the face-to-face and online teaching environments, with a focus on high-value contact time."

(2)    "Changing transport patterns defined the 'new normal', shifting from the major flow of public to and from city centres in the peak hours (Monday to Friday), towards fewer, but more frequent journeys throughout the day to regional centres."

(3)    "The understanding of the resident population of the city centre changed to be more flexible and seasonal than before. Some opportunities, such as sourcing co-working hubs within local neighbourhoods, were introduced as alternatives for commuting to city centres on a daily basis."

The opportunities to repurpose buildings and introduce multipurpose spaces were recognised across all communities.

(1)    "An attractive city centre would entail one that offers great flexibility and options, inclusive of communal meeting spaces, and design elements [on campus] which prioritise public health."

(2)    "Adequate public transport provision, flexibility, and options to support a safe environment will create an attractive city centre."

(3)    "Active transport modes, efficient links between the city centre and regional hubs, fast connections, flexible movement patterns, and prioritization of the health agenda through planning and design elements will play an increasingly important role in terms of attractiveness of the urban-scape."

The opportunity to embrace "active transport" modes became more prominent, in response to the health agenda and congested passive transport routes.

(2)    "The public sector is revisiting the notion of public goods from a health perspective. For Transport for NSW it implies reflecting on concepts of 'active working' and 'active transport' and how to align the public sector infrastructure provision with fast-changing social needs."

(3)    "The use public transport was inhibited by the fear of COVID-19 infection and challenges in enforcing social distancing regulations. The use of private transportation modes, and active mobility such as walking or cycling to travel to and from the workplace, significantly increased since the start of the COVID-19 pandemic."

The creation of working hubs and activity-based working spaces and possibilities were also highlighted as opportunities for the future urban-scape.

(3)    "While city centres will continue to be the hubs for business, the permanent nature of these will change and make room for more flexible working arrangements with temporary (rotating) staff, leaving many of the envisioned office spaces unoccupied. Some of these commercial spaces could be repurposed to become accommodation for workers or to accommodate multipurpose uses. It does imply a vast shift in thinking about inward investment and national and international capital for developments in city centres. A 'new normal' for city centres is expected where businesses will operate from remote locations (regional or suburban) to accommodate flexibility. This would imply (even if temporarily) some opportunity to convert offices and to repurpose these spaces for accommodation or other uses."

The COVID-scape redefined the broader understanding of 'business as usual' and has introduced various challenges and opportunities to be considered for planning liveable (future) city centres. The reflections highlighted the prominence of the human-scape as a critical factor guiding the urban-scape, and the changing trends and evolving multiculturism that were brought along by the recent pandemic, of which some might be part of the 'new normal' going forward.

## 6. Discussion: Transitioning from Urban-Scape to Human-Scape

The urban-scape will always be impacted by the knowledge production, service provision, productivity, innovation, and economic development [41] that contribute to the agglomeration forces which define the city centre, but human-scape considerations such as high-value connection spaces to embrace quality of life, deep connections, cultural recharge,

and sanctuary will progressively take the lead in shaping future cities. Current trends at this time predict that social considerations will far outweigh the economic competitiveness of spaces and will increasingly be the core denominator in future city centres where the transition from an urban-scape to a fit-for-purpose human-scape will be evident. This would, however, imply a change to 'business as usual', where greater flexibility, choice, and social needs would lead the conversation on the attractiveness of city centres. Based on the reflections included in this research, the following themes were identified as the core issues that will support the transition from contemporary, post-COVID urban-scape to human-scape.

### 6.1. Changing Movement Patterns

The global uptake in remote working would possibly continue in the future even when, to a lesser extent, it will have severe impacts in terms of the broader movement patterns in (and around) cities. The decline in public transport patronage was evident as movement patterns changed according to the societal needs and 'active transport' options saw great interest, confirming previous research. A decline of the resident population of the city centre was evident, both in terms of local users of the space (the regulars) visiting the city centre daily or weekly, and also in terms of visitors (tourists) who used to shape the city centre. Flexible movement patterns and fast (selective) connections would define future cities in supporting the transition from the urban-scape to the human-scape.

### 6.2. Changing Social Infrastructure

There is a new understanding of "available versus accessible". As remote working and flexible working hours are becoming more prominent, the need for supporting social infrastructure will likewise also expand. The rapid deployment of collaborative tools and technologies will define our future cities to enable communities to work effectively and efficiently. City centres will become an integral part of the broader ecosystem as we progressively understand that economic prosperity depends upon healthy social structures. This research confirms that economic prosperity will depend upon healthy social structures. The significance of urban identity would need to be revisited in light of sense of place theories and quality of life objectives, pertaining to the dimensions set forward by Lynch (1981) [9] that contemplate how form will affect vitality and human sense, how people could better access the activities and services of the city centre, and how societal needs will shape these urban spaces. New advances in communication technologies and supporting urban metropolitan transport systems now allow citizens to stay selectively in touch, while disconnecting from the city at large, reframing the social infrastructural needs of traditional city planning approaches. Social infrastructure will play a critical part in future cities as the enabler of social connections, bringing societies together for more fractional, but high-value interactions.

### 6.3. Increasing Multifunctionality

During the first quarter of 2021, the estimated occupancy levels in the city centre of Sydney were between 25% and 40%. Occupancy levels were, however, drastically reduced after the June 2021 lockdown, with some experts questioning if this would entail a more 'permanent reminder of the new normal'. The evidence that urban space is changing calls for a renewed understanding of urban land value and multifunctionality, where specific spaces could be better used to accommodate a range of uses, users, and activities. As a result, the urban identity now has to incorporate the new dynamics of a COVID generation, and multifunctionality will be increasingly employed in future cities to accommodate changing needs, activities, and forms of interaction, ultimately accommodating the change from urban-scape to human-scape. Table 3 captures the trends identified from the urban-scape in accordance with the human-scape considerations that will ultimately shape attractive (future) city centres.

**Table 3.** Changing trends that will define future cities.

| Issues | Trends Identified from the Urban-Scape | Human-Scape Considerations to Shape Attractive (Future) City Centres |
|---|---|---|
| Changing movement patterns | Greater flexibility brought about by the impact of remote working policies on the function and form of the urban-scape. A disconnection from the urban-scape was evident, while a deeper connection to the human-scape arose. | • Prioritise the planning of "active transport" modes<br>• Plan for activity-based environments<br>• Accommodate flexible uses, users, and activities as part of city planning approaches<br>• Include transport systems that allow for fast, flexible connections to the urban-scape<br>• Plan for temporary spaces |
| Changing social infrastructure | Hybrid working ways changed the infrastructural needs and interaction between 'availability' and 'accessibility'. | • High-value connection spaces should support accessibility and availability<br>• Urban spaces should be planned for enhanced experience and meaningful encounters<br>• Availability of staff should not be location-focused, rather supported by technologies to bridge geographical distances<br>• Social considerations should lead the notion of attractiveness of city centres |
| Increasing multifunctionality | The change in urban land use was observed on a global scale. The need for multifunctional spaces to accommodate changing activities, users, and uses were continuously highlighted. | • Healthy environments should be prioritised as part of broader spatial planning approaches<br>• Focus should be placed on the greater deployment of collaboration technologies<br>• The end goal should be to enhance user experience within the city centre |

### 7. Conclusions: Shaping Attractive Future City Centres

The human-scape, contextualised by social equity and social capital, will become more prominent in the future, addressing a universal fulfilment of the most fundamental human needs within the urban-scape [12,42,43]. The urban identity will accordingly be influenced by user-values and user-needs, driven by human capital, social issues, and liveability considerations. The future of the city centre is dependent on our ability to successfully manage the transition from an urban-scape to a human-scape. It implies the human-scape being articulated in broader spatial planning approaches to respond to the impacts of changing movement patterns, changing social infrastructure, and increasing multifunctionality as core themes. While these impacts challenged the contemporary urban-scape, they also provided a unique opportunity to shape future cities following the trends that were set in motion during the COVID-19 pandemic. Active transport modes, greater flexibility, temporary uses and spaces, improved accessibility, enhanced experiences, and healthy environmental and collaborative technologies will define the urban identity of future city centres. In a quest to reclaim attractive city centres, the urban-scape would need to transition to embrace and prioritise the human-scape.

**Author Contributions:** Conceptualization, E.J.C. and S.S. All authors contributed to the investigation, writing, review and editing. All authors have read and agreed to the published version of the manuscript.

**Funding:** This research was supported by the National Research Foundation South Africa, Grant No. 116243.

**Informed Consent Statement:** Informed consent was obtained from all subjects involved in the study.

**Data Availability Statement:** Not applicable.

**Conflicts of Interest:** The authors declare no conflict of interest.

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
