# Peer review of "From Urban-Scape to Human-Scape: COVID-19 Trends That will Shape Future City Centres"

_land, doi:10.3390/land10101038_

Round 1
Reviewer 1 Report
The paper "From urban-scape to human-scape: What we need to know to create competitive and attractive future city centres", through content and approach is extremely interesting. It is imposed as a necessity, against the background of the mutations that define the current society and in the context of the Covid pandemic 19. The use of a specific narrative analysis methodology to analyze different human-landscape perspectives on the impact of COVID-19 on cities, led to results, conclusions and recommendations, which, in one way or another, we all feel and intuit. However, I recommend improving the study using studies with similar topics from the literature in the last period of time (2019 -2021). Given the topicality of the research, the methodology used, the logical sequence and the results obtained support the publication of the article entitled "From urban-scape to human-scape: What we need to know to create competitive and attractive future city centres" in its current form, with minor modifications, form imposed by the requirements of the magazine.
Also on this occasion, I would like to congratulate the research team for their concerns and achievements. Luck!
Author Response
Thank you for the valuable feedback. We have revised the paper accordingly and included the following changes:
- More recent literature has been included in the review section, specifically articles from 2019-2021 relating to different facets of cities after the pandemic began (over 12 new articles inc.), and which relate to the communities in the reflections.
- We increased the emphasis on the two scapes (human and urban scape) by re-contextualising/re-writing introductory sentences and being more explicit about which 'scape' is under discussion in the paragraph. Economic emphasis has been reduced to focus more about the shift away from city centres as economic hubs to the wider functions of cities today.
- Key concepts have been added, eg: to define concepts of scape adopted by the paper
- Focus of the theoretical section, has been restructured and content added to discuss the key ideas/changes in recent literature in the human and urban scapes, relevant to the 3 communities in the analysis & results sections.
- The explanation of use of expert contributions as part of the empirical investigation is now included.
- The conclusions and recommendations sections were revised to better accentuate the findings and specific drivers that were identified and proposed for the planning of future city centers.
- A native English speaking person reviewed the text, however, due to time constraints we did not have the paper professionally edited. We are still willing to do this if needed, as well as capture the references in numerical order.
Reviewer 2 Report
Dear authors,
The concerns over the current adaptations of cities’ due to Covid-19 pandemic are obvious on the agenda of most (all) city officials and urban planners. The scientific contributions to this change are, thus, most welcome. I will detail below my main comments regarding your text. Hope the authors may seem them as constructive and supportive for the revision of the text.
Major comments:
- The change from urban-scape to human-scape is key to your thesis. However, when reading the text, it seems to be an almost full-centered concern over the economic dimension. Often, the analysis of the individuals is to discuss how it may affect the city and its CBD economic performance/competitiveness. Having said this, your use of sustainability or the term ecosystem is fruitless, as it is not applied in the analysis, nor in the discussion.
- The theoretical section is vague and does not explore any key concept. This is a major setback, as it gives the impression that the authors were tempted to move too quickly into the empirical section, without establishing a conceptual framework. The text, in its current form, does not add much to the literature. Mostly because there is the lack of a robust theoretical section, but also consubstantiated by the fact that most quotes are not particularly new to anyone informed about the impacts of the pandemic in our cities.
- As mentioned regarding the theoretical section, the recommendation section is too vague. It is important that the authors may be willing to provide more accurate and in-depth insights.
Minor comments
- I’m personally not an English native, but some typos are possible to be found along the text. See for instance the following lines:11, 12, 57, 72, 116, 136, 271, 394, 433.
- I do not agree on the need for a sub-section 4.3. This sub-section only has 5 lines, it can be easily incorporated in the previous one.
Line 484: The incorporation of “communications” must be a typo.
- How many interviews were performed. There is the placement of a significant set of quotations, but they were extracted from how many interviews; and what are these interviewees positions (exception made to the three co-authors already mentioned in the respective section).
Author Response

(The authors gave the same response as above.)

Round 2
Reviewer 2 Report
Dear authors,
Appreciate the efforts to improve the text. However, the text is very similar to the initial version and the changes were not in-depth, at least not in the extent I believe it was necessary to truly improve the article, for reasons I will further detail. In general terms, my major comments of the first version are still applicable to this new revised version.
- This section is very similar to the one of the previous version. The part of the text that was further changed – regarding the adding of references mentioned by the authors – is on section 3, which is an introduction to the empirical section. I’m not questioning the validity of what was done, in particular because it must be acknowledged that it was to comply with the need to introduce some text on non-economic scapes, but the article would better engage with other literature on the scientific field if a specific literature review section was established.
- It still remains to address my comment on the need to clarify the issues over the interviews. To be clear, I believe it is important to clarify how many people were interviewed in each institution; when it was done; in which mode (in person or other).
- Contrarily to what is mentioned, there is no improvement to the recommendation sub-section. The authors just choose to retrieve the first two paragraphs (including the sub-title).
- Some English editing or further revision on the part of the authors is advised. I must enhance again that, despite, not being an English native, some errors persist in this version. The text is very readable, thus, probably another check from the authors will solve this issue.
Author Response
The following revisions were made:
- Revised the title of the paper to better match the content
- Improved the flow and content of section 4.
- Added some linking sentences before & after the quotes.
- Methodology reviewed to state that the research considered reflections of the impact of COVID-19 on the city from three different perspectives: 1) the community sector, 2) the public sector, and 3) the private sector. The perspectives were captured by means of expert contributions, of individual stakeholders sharing their reflections from each of the respective sectors. Supporting literature for including expert reflections were also included.
- Changed the titles of sections 4.1, 4.2, & 4.3 to match the content more closely and make the sections more distinct
- Revised section 5 in total to highlight changing trends based on the reflections provided in the paper
- Table 3 was included as a revised version, showing the trends identified from the urban-scape in accordance with the human-scape considerations that will shape attractive (future) city centres.
- Revised conclusions and main findings in totality.
- Language editing and technical editing completed.
